# Development and Characterization of a Cancer Cachexia Rat Model Transplanted with Cells of the Rat Lung Adenocarcinoma Cell Line Sato Lung Cancer (SLC)

**DOI:** 10.3390/biomedicines11102824

**Published:** 2023-10-18

**Authors:** Eiji Kasumi, Miku Chiba, Yoshie Kuzumaki, Hiroyuki Kuzuoka, Norifumi Sato, Banyu Takahashi

**Affiliations:** R&D Laboratories, EN Otsuka Pharmaceutical Co., Ltd., Hanamaki 025-0312, Japansato.norifumi.a@otsuka.jp (N.S.);

**Keywords:** cancer cachexia, animal model, anorexia, tube feeding, SLC cell line, weight and muscle losses, NGAL

## Abstract

Cancer cachexia is a complex malnutrition syndrome that causes progressive dysfunction. This syndrome is accompanied by protein and energy losses caused by reduced nutrient intake and the development of metabolic disorders. As many as 80% of patients with advanced cancer develop cancer cachexia; however, an effective targeted treatment remains to be developed. In this study, we developed a novel rat model that mimics the human pathology during cancer cachexia to elucidate the mechanism underlying the onset and progression of this syndrome. We subcutaneously transplanted rats with SLC cells, a rat lung adenocarcinoma cell line, and evaluated the rats’ pathophysiological characteristics. To ensure that our observations were not attributable to simple starvation, we evaluated the characteristics under tube feeding. We observed that SLC-transplanted rats exhibited severe anorexia, weight loss, muscle atrophy, and weakness. Furthermore, they showed obvious signs of cachexia, such as anemia, inflammation, and low serum albumin. The rats also exhibited weight and muscle losses despite sufficient nutrition delivered by tube feeding. Our novel cancer cachexia rat model is a promising tool to elucidate the pathogenesis of cancer cachexia and to conduct further research on the development of treatments and supportive care for patients with this disease.

## 1. Introduction

Cachexia is an abnormal metabolic disorder that is characterized by muscle mass loss and is often associated with anorexia, inflammation, insulin resistance, fatigue, and weakness [1]. Cachexia affects the prognosis and patients’ quality of life in the late stages of serious diseases, such as cancer, acquired immunodeficiency syndrome, chronic obstructive pulmonary disease, kidney disease, rheumatoid arthritis, neurological disease, and heart failure [2,3].

Cancer cachexia is defined as “a multifactorial syndrome defined by an ongoing loss of skeletal muscle mass (with or without the loss of fat mass)” [4], and it affects 50–80% of patients with solid cancers [5]. In addition to the typical symptoms, such as weight loss and anorexia, cancer cachexia drastically decreases a patient’s quality of life and causes poor chemotherapeutic responses, increases toxicity, interrupts treatments, and reduces patient survival time [6,7]. While cancer cachexia causes malnutrition-like symptoms, such as loss of body weight and reduction in food intake, it differs from starvation and simple malnutrition, both of which can be easily treated by providing appropriate nutrients. Studies have reported that supplying energy alone to patients with cancer cachexia does not increase their body weight, as demonstrated by the ineffectiveness of conventional nutritional interventions [8,9].

Abnormal muscle and fat metabolism are widely accepted as crucial factors that affect the pathogenesis of cancer cachexia. In fact, patients with cancer cachexia have increased resting energy expenditure, increased protein degradation and lipolysis, and decreased protein synthesis compared with healthy individuals [10]. These metabolic changes are caused by an increase in the levels of tumor-specific local and systemic inflammatory mediators by tumor and/or host immune cells. For instance, protein degradation is mediated by the ubiquitin–proteasomal system comprising the muscle-specific ubiquitin ligases, including muscle-specific ring finger protein 1 (MuRF1) and atrogin-1 [11]. Most proinflammatory cytokines regulate protein metabolism in the muscles via several intracellular signal transduction pathways. Moreover, these cytokines convert white adipose tissue into brown adipose tissue and enhance mitochondrial thermogenesis by increasing the expression of uncoupling protein 1. As a result, increases in lipid mobilization and resting energy expenditure are observed [12].

Anorexia is another important factor associated with cancer cachexia because reduced food intake and metabolic abnormalities in a majority of patients drive weight loss in cancer cachexia. Anorexia in cancer is common in the advanced stages of several cancers and is often exacerbated by treatment-induced adverse effects (e.g., nausea, vomiting, and oral mucosal ulcers), decreased motor activity, tumor-related gastrointestinal obstruction, and psychological factors [13]. Proinflammatory cytokines induce both muscle atrophy and lipid mobilization [14]. Furthermore, these compounds increase corticotropin-releasing hormone production and suppress gastrointestinal motility, thereby decreasing appetite [15]. These cytokines also transmit leptin-like signals to the hypothalamus, resulting in loss of appetite and inducing body weight control responses that mimic the condition of sufficient body fat reserves [16]. Patients with certain types of cancer cachexia have elevated plasma levels of ghrelin (the only appetite-enhancing peptide secreted by the stomach). However, these patients still suffer anorexia; thus, it can be deduced that ghrelin receptor sensitivity may be reduced in this disease [17].

Despite the increasing development of new drugs that are anti-inflammatory and enhance patient appetite and protein synthesis in cancer cachexia, these drugs are yet to be approved for clinical use [18,19,20]. This may be because several inter-related factors form the basis of this disease; however, only a few rodent models adequately reflect the pathological state of human cancer cachexia. Here, we present a promising novel rodent model of cancer cachexia that closely resembles the pathology of human cancer cachexia, particularly that of severe anorexia, to elucidate the complex etiology of this disease.

## 2. Materials and Methods

### 2.1. Animals

All animal protocols were conducted in accordance with regulations for experiments involving laboratory animals, established by the Animal Use and Care Committee of EN Otsuka Pharmaceutical Co., Ltd., Hanamaki, Japan (approval no. ENDR-15-02, 29 May 2015). We housed 4-week-old male F344/NJcl.Cg-Foxn1rnu rats (CLEA Japan, Inc., Tokyo, Japan) in individual stainless steel metabolic cages under controlled temperature (23 ± 3 °C) and humidity (50 ± 20%) conditions and in a 12 h/12 h light/dark cycle (lights on at 07:00). The rats were fed AIN-93G (Oriental Yeast Co., Ltd., Tokyo, Japan) and provided with drinking water ad libitum. Additionally, the rats were allowed to acclimatize to the laboratory conditions for approximately 1 week before the experiments. The rats were randomly assigned to control or SLC transplantation groups, pair-matched on the basis of body weight.

### 2.2. Cell Culture and Tumor Inoculation

We purchased the SLC cell line from RIKEN BioResource Research Center (Tsukuba, Japan; Cell No.: RCB2862) and cultured the cells, in accordance with product sheet recommendations from the RIKEN BioResource Research Center, in RPMI-1640 medium (Nacalai Tesque, Inc., Kyoto, Japan). This medium was supplemented with 10% fetal bovine serum, 2 mmol/L L-glutamine, 100 U/mL penicillin, and 100 µg/mL streptomycin and the cells were cultured in 5% carbon dioxide at 37 °C. The cells were harvested from 80% confluent cultures using Accutase™ (Nacalai Tesque, Inc., Kyoto, Japan) and suspended at a concentration of 5 × 10^5^ cells/mL in phosphate-buffered saline. The rats were anesthetized with isoflurane (Pfizer Inc., New York, NY, USA) and subcutaneously inoculated in the right flank with 0.1 mL of the cell suspension using a 26 G needle (Terumo Corporation, Tokyo, Japan).

### 2.3. Grip Strength Test

We quantified the grip strengths of the rats’ forelimbs using a grip-strength meter (Bioseb, Vitrolles, France). In accordance with the manufacturers’ instructions, we lifted the rats by their tails to allow their front paws to grab a measuring bar. Subsequently, we pulled them horizontally backward by their tails until they released their front paws. The maximum force applied by the front paws to grasp the bar was recorded. This test was performed in triplicate for each rat, and the highest and lowest values were discarded; the remaining value was used for the subsequent analysis.

### 2.4. Open Field Test

We also tested the locomotor movements of the rats in an open field (45 × 45 × 20 cm; Infrared Actimeter, Panlab Harvard Apparatus, Barcelona, Spain). Both horizontal and vertical movements of the rats were measured by counting the number of infrared beam interruptions that were recorded. We collected these data for 10 min and analyzed the data using ActiTrack software version 2.7 (Panlab Harvard Apparatus, Barcelona, Spain). The following parameters were calculated: distance travelled: total distance travelled; locomotor activity: number of times the infrared beam sensor responded; and resting time: the total amount of time the infrared beam sensor was unresponsive.

### 2.5. Blood Measurements

At autopsy, we collected 6 mL blood samples (for enzyme-linked immunosorbent assay, blood biochemistry, and hematology analyses) from the venae cava of isoflurane-anesthetized rats in ethylenediaminetetraacetic acid-coated tubes before euthanasia. We analyzed the clinical chemistry of each blood sample by centrifuging the sample at 1500× *g* for 15 min and collecting the resultant serum. Hematological assessment and serum chemistry were evaluated using a hematological analyzer, XT-1800iV (Sysmex Corp., Hyogo, Japan), and a clinical chemistry analyzer, Fuji DRI-CHEM 3500V (FUJIFILM Medical Co., Ltd., Tokyo, Japan), respectively, in accordance with the manufacturers’ instructions. We determined the interleukin-1 beta (IL-1β) and parathyroid hormone-related peptide (PTHrP) concentrations using rat IL-1β Quantikine (R&D Systems, Inc., Minneapolis, MN, USA) and a rat PTHrP enzyme-linked immunosorbent assay kit (MyBioSource, Inc., San Diego, CA, USA), respectively, in accordance with the manufacturers’ instructions.

### 2.6. Cytokine Array

We quantified the cytokine levels in the pooled rat serum, cell culture supernatants, and tumor extracts using the Proteome Profiler Rat Adipokine Array Kit (R&D Systems, Inc., Minneapolis, MN, USA) in accordance with the manufacturer’s instructions. To perform this analysis, we pooled an equal volume of serum from each rat. We then homogenized rat tumors in phosphate-buffered saline with a protease inhibitor cocktail (Nacalai Tesque, Inc., Kyoto, Japan) and 0.05% Triton-X (Sigma-Aldrich Japan, Tokyo, Japan). The homogenates were centrifuged at 20,000× *g* for 15 min at 4 °C, and the supernatants were used as tumor extracts. We mixed the pooled rat serum, cell culture supernatants, and tumor extracts with a cocktail of biotinylated detection antibodies. This mixture was incubated with a nitrocellulose membrane spotted with capture antibodies. Subsequently, we added streptavidin–horseradish peroxidase and chemiluminescent detection reagents to the membrane. The chemiluminescent signals from spots on the membrane were captured (LAS 4000; GE Healthcare Japan Corporation, Tokyo, Japan) and quantified using Image Gauge version 4.0 (GE Healthcare Japan Corporation, Tokyo, Japan).

### 2.7. RNA Extraction, Reverse Transcriptase Polymerase Chain Reaction (PCR), and mRNA Quantification

For mRNA quantification, we extracted the RNA from the rats’ skeletal muscle tissues using ISOGEN (FUJIFILM Wako Pure Chemical Corporation, Osaka, Japan) in accordance with the manufacturer’s instructions. We then synthesized complementary DNA from the extracted RNA using Prime Script Transcriptase (Takara Bio, Inc., Shiga, Japan). Next, we performed semi-quantitative reverse transcriptase PCR testing using the synthesized complementary DNA as the template. mRNA-specific fragments were identified using linear-phase PCR amplification; the mRNA expression levels of the target proteins were normalized with that of glyceraldehyde-3-phosphate dehydrogenase. The gene-specific primer pairs used in this study are provided in Appendix A.

### 2.8. Tube Feeding following Gastrostomy

The rats underwent gastrostomies as described previously [21]. Briefly, the rats were anesthetized with isoflurane (Pfizer) following an overnight fast. Next, we performed laparotomies and inserted catheters (Terumo Corporation, Tokyo, Japan) into the rats’ stomachs. The distal ends of the catheters were tunneled subcutaneously through the lateral abdominal walls and exited at the dorsal neck. Each rat received an antibiotic (penicillin G potassium; Meiji Seika Pharma Co., Ltd., Tokyo, Japan) intraperitoneally before abdominal closure. The rats were allowed to move freely within their individual cages after recovery from surgery, as the proximal end of the catheter was attached to a swivel spring. The rats were given a liquid diet (Appendix A), which was prepared at EN Otsuka Pharmaceutical Co., Ltd. (Iwate, Japan), through an infusion pump (SP-115; JMS Co., Ltd., Tokyo, Japan). Depending on the rats’ postoperative conditions, we gradually increased the dose of the liquid diet over time.

### 2.9. Histological Analysis of the Skeletal Muscles

Following euthanasia, we isolated the rats’ skeletal muscles and fixed them in 10% buffered formalin (FUJIFILM Wako Pure Chemical Corporation, Osaka, Japan) for over 24 h. Subsequently, the muscles were injected stepwise with 30%, 20%, and 10% sucrose solutions and kept overnight for cryoprotection. The muscle tissues were embedded in optimal cutting temperature compound (Sakura Finetek Japan Co., Ltd., Tokyo, Japan) and flash-frozen in liquid nitrogen. Subsequently, we stained 10 µm cryosections of the rat skeletal muscle tissues with hematoxylin and eosin. The cross-sectional area (CSA) and boundary lengths of the gastrocnemius muscle fibers were analyzed using analySIS FIVE software version 1.9 (Olympus Corporation, Tokyo, Japan).

### 2.10. Statistical Analysis

The data are expressed as mean ± standard deviation. Data were analyzed statistically using Student’s *t*-test or Welch’s *t*-test on the basis of the result of an F test (GraphPad Prism 9 version 9.5.1; GraphPad Software, San Diego, CA, USA). Values of *p* < 0.05 were considered statistically significant.

## 3. Results

### 3.1. SLC-Transplanted Rats Exhibited Severe Weight Loss and Reduced Food Intake

Initially, we measured the rats’ body weight and food intake to investigate the effects of subcutaneous SLC transplantation. We observed that SLC-transplanted rats had dramatically lower body weights compared with the control rats approximately 15 days after transplantation (Figure 1A; *p* < 0.01). In particular, we recorded a remarkable decrease in food intake in the SLC-transplanted rats. Moreover, the energy intake of these rats decreased to approximately 10% of the intake of the control rats on day 17 following transplantation (Figure 1A; *p* < 0.01). The substantial weight loss of the SLC-transplanted rats resulted in an extremely thin appearance, with conspicuous ribs (Figure 1B). Subsequently, we performed necropsies to examine whether this weight loss was accompanied by muscle and/or fat loss. We observed that the muscle mass in the legs of the SLC-transplanted rats was considerably reduced macroscopically compared with that of the control rats (Figure 1C). Notably, the SLC-transplanted rats exhibited significant muscle loss in all examined skeletal muscles, including those on the carcass, and the gastrocnemius, soleus, and quadriceps, and adipose tissue loss, including epididymal, perinephric, and mesenteric adipose tissues (Figure 1D,E; *p* < 0.01). Thus, the physical characteristics of the rats in our rat model were quite similar to the typical symptoms of people with cancer cachexia.

### 3.2. SLC Transplantation Induced Muscle Fiber Atrophy and Upregulated the Expression of Muscle-Specific E3 Ubiquitin Ligases in Rats

We histologically examined the CSAs and boundary lengths of the gastrocnemius muscle fibers to estimate the severity of the rats’ muscle loss. The SLC-transplanted rats had significantly lower CSAs and boundary lengths compared with the control rats (Figure 2A; *p* < 0.01). Moreover, we measured the mRNA levels of muscle-specific degradation factors associated with muscle atrophy in cancer cachexia [11]. In fact, we documented that the levels of the muscle-specific ubiquitin ligases MuRF1 and atrogin-1 were markedly upregulated in the gastrocnemius muscles of the SLC-transplanted rats compared with levels in the control rats (Figure 2B; *p* < 0.01).

### 3.3. Muscle Atrophy Associated with SLC Transplantation Reduced Muscle Function and Activity in Rats

We further explored whether the muscle weight loss and muscle atrophy affected muscle function in SLC-transplanted rats. For this purpose, we performed behavioral tests because decreased physical function has been associated with decreased muscle weight in patients with cancer cachexia [22]. The grip strength test, which estimated the muscle strength of the rats’ forelimbs on the day of necropsy, revealed that SLC-transplanted rats had a significantly lower grip strength compared with the control rats (Figure 3A; *p* < 0.01). We then examined the locomotor activities of the rats by performing an open field test. The distances traveled by the SLC-transplanted rats per measurement of time were extremely low compared with those of the control rats (Figure 3B,C; *p* < 0.01). The locomotive activities of the SLC-transplanted rats were also significantly lower than those of the control rats (*p* < 0.01). Of note, approximately half of the measured time for the SLC-transplanted rats was resting time. Thus, our data clearly demonstrated that SLC transplantation reduced both muscle weight and function in the rats by increasing the mRNA expression levels of the muscle-specific degradation factors (E3 ubiquitin ligases).

### 3.4. SLC-Transplanted Rats Developed Severe Anemia, Malnutrition, and Hypercalcemia

Studies have reported that patients with cancer cachexia exhibit symptoms of anemia and malnutrition resembling symptoms of hypoalbuminemia, in addition to anorexia and weight loss [23]. Thus, we evaluated whether the SLC-transplanted rats exhibited these symptoms by performing hematological assessments and analyzing the rats’ blood biochemistry. The SLC-transplanted rats displayed severe anemia and had significantly low red blood cell and platelet counts and hemoglobin and hematocrit levels (Figure 4A; *p* < 0.01). Our serum biochemistry data revealed that the blood albumin and total protein levels in the SLC-transplanted rats were lower than those in the control rats (Figure 4B). Notably, the serum calcium concentration was extremely high in the SLC-transplanted rats (hypercalcemia). Cancer-associated anemia is a proinflammatory cytokine-mediated disorder that occurs because of complex interactions between tumor cells and the immune system [24]. Therefore, we subsequently examined the inflammatory status of the SLC-transplanted rats. The inflammatory cytokine IL-1β, which has been implicated in cancer-associated anemia and cachexia through hypermetabolism [25], was not detected in the sera of the control rats. However, IL-1β was elevated in sera of the SLC-transplanted rats (Figure 4C). Furthermore, hypercalcemia, observed in the SLC-transplanted rats, is a common paraneoplastic symptom and is induced by PTHrP. As studies have reported that PTHrP is a causative factor of cancer cachexia, we measured the PTHrP level in the rats’ blood samples [26]. The PTHrP levels were significantly higher in the SLC-transplanted rats compared with the control rats (Figure 4D). Thus, our data suggested that SLC-transplanted rats and patients with cancer cachexia both exhibit weight loss, anemia, and malnutrition. Moreover, we deduced that these symptoms are induced via inflammation and PTHrP.

### 3.5. The Cytokine Production Patterns in SLC-Transplanted Rats Were Significantly Different from Those in the Control Rats

We further investigated the detailed mechanism underlying SLC transplantation-induced cancer cachexia and characterized the SLC-transplanted rat model. We performed a cytokine array to identify the humoral factors that are likely involved in cancer cachexia (Figure 5A). The SLC-transplanted rats had >2-fold higher serum levels of fibroblast growth factor 21 (FGF21), insulin-like growth factor-binding protein 3, IL-1β, IL-10, neutrophil gelatinase-associated lipocalin (NGAL), plasminogen activator inhibitor type 1 (PAI-1), tumor necrosis factor alpha (TNF-α), and vascular endothelial growth factor than those of the control rats (Figure 5B). In particular, the production levels of NGAL and PAI-1 differed considerably between the sera of the control and SLC-transplanted rats. In contrast, the serum levels of IGF-1, IL-11, and leptin were >2-fold lower in the SLC-transplanted rats than those of the control rats. Furthermore, we detected intercellular adhesion molecule 1, insulin-like growth factor-binding protein 6, IL-11, leukemia inhibitory factor (LIF), NGAL, monocyte chemoattractant protein 1, macrophage colony-stimulating factor, PAI-1, tissue inhibitors of metalloproteinases 1, and vascular endothelial growth factor levels in the culture supernatants of the SLC cell line. Remarkably, the cytokine production patterns were similar between the tumor extracts and the SLC-transplanted rat blood samples (Figure 5C). Thus, the cytokine array helped identify the serum cytokines that were likely involved in the pathology of cancer cachexia in the SLC-transplanted rats.

### 3.6. SLC Transplantation Induced Muscle Atrophy and Loss of Muscle Function in Rats Subjected to Tube Feeding

We determined that the cancer cachexia symptoms, such as weight loss, muscle atrophy and anemia, induced by SLC transplantation were not solely caused by simple starvation attributed to reduced food intake. Therefore, we investigated the effects of SLC transplantation by tube feeding a liquid diet using a gastrostomy catheter (Figure 6A). As shown in Figure 6B,C, the SLC-transplanted rats had significantly lower body and muscle weights compared with the control rats, even when all rats were fed an isocaloric diet. However, interestingly, we observed that the adipose tissue weights did not decrease in the SLC-transplanted rats fed an isocaloric diet (Figure 6D). The CSAs and boundary lengths of the muscle fibers and the grip strengths in the SLC-transplanted rats displayed similar patterns of decrease. In contrast, the expression levels of muscle-specific ubiquitin ligases in the gastrocnemius muscles of the force-fed SLC-transplanted rats increased significantly over time (Figure 6E–G). Notably, the locomotor activities of the force-fed SLC-transplanted rats decreased over time (Figure 6H). Additionally, these force-fed rats were anemic and malnourished and had low serum albumin and total protein levels (Figure 6I,J). These data clearly demonstrated that the cancer cachexia symptoms induced by SLC transplantation were not because of simple starvation alone.

## 4. Discussion

In the present study, we demonstrated that rats subcutaneously transplanted with SLC cells displayed weight loss, muscle atrophy, severe anorexia, anemia, elevated inflammatory marker levels, and decreased serum albumin levels. Evans et al. reported that in addition to weight loss, cachexia is associated with three of the following five symptoms: reduced grip strength, fatigue, low energy intake, low muscle mass, and abnormal biochemical parameters (such as increased inflammatory marker levels, anemia, and low albumin levels) [1]. Consistent with this definition, the rats in our SLC-transplanted model exhibited all of these symptoms except fatigue. Although it is difficult to evaluate the degree of fatigue in a laboratory animal, several studies have reported that increased fatigue is accompanied by a decrease in spontaneous activity when using sensors to measure activity levels [27]. While the possibility of muscle weakness because of the loss of muscle mass cannot be denied, the decrease in the locomotor activity of the rats in our model (measured by the open field test) may imply an increase in fatigue in the SLC-transplanted rats. Additionally, the definition of cachexia in our model concurred with the cancer cachexia-specific definition advocated by Fearon et al. in 2006, which states that cancer cachexia is associated with weight loss, decreased food intake, and systemic inflammation [28]. Importantly, the current consensus definition states that conventional nutritional therapies cannot treat cancer cachexia [4]. Indeed, clinically, nutritional interventions, such as parenteral and enteral nutrition, in patients with cancer cachexia with reduced dietary intake barely improve or fail to improve the patients’ nutritional status [29]. Furthermore, the effects of nutritional intervention in several cancer cachexia animal models with reduced food intake have not been verified. Therefore, this is the first study to our knowledge that demonstrates the ineffectiveness of nutritional intervention in a cancer cachexia animal model involving tube feeding an isocaloric liquid diet, reflecting the human pathology. 

Similar to patients with cancer cachexia, the rats in our model exhibited barely any improvement in symptoms, particularly nutritional status, even when eating an isocaloric diet. Furthermore, the SLC-transplanted rats exhibited similar degrees of muscle loss in both the tube-feeding and free intake experiments. In contrast, the adipose tissue weights of the SLC-transplanted rats did not differ from those in the control rats when both were tube-fed. Thus, we deduced that the decrease in muscle mass in the rats in our cancer cachexia model was a direct effect of tumor-secreted humoral factors or tumor–host interactions, whereas fat loss is an indirect effect of decreased dietary intake because of anorexia; we also deduced that the weight loss appeared as a result of both effects. Clinically, nutritional support by parenteral or enteral feeding has been demonstrated to only increase fat mass in patients with cancer cachexia [30,31], thereby supporting the similarity of our model to the human cancer cachexia pathology. With these considerations, our results clearly indicate that our SLC-transplanted cancer cachexia rat model efficiently mimics the pathophysiology of cancer cachexia patients.

In our study, a substantial reduction in food intake, similar to that in severe anorexia in people, was quite characteristic to our cancer cachexia rat model. In fact, we identified several humoral factors that could be related to the altered appetite of SLC-transplanted rats by enzyme-linked immunosorbent assays and cytokine arrays. Notably, PTHrP suppresses food intake and delays gastric emptying by increasing the levels of hypothalamic urocortin 2 and 3. Moreover, it is well established that patients with PTHrP-induced hypercalcemia are severely anorexic [32]. Recently, studies have revealed that FGF21, the levels of which were increased in the serum of the SLC-transplanted rats, regulates a person’s appetite [33]. FGF21 is a hormone secreted by the liver that exhibits a wide range of metabolic effects via the co-receptors FGF receptor 1 and β-Klotho. FGF21 also activates the paraventricular nucleus of the nesfatin-1 neurons by crossing the blood–brain barrier to regulate food intake in laboratory animals [34,35]. Moreover, when mammals starve, the blood concentration of the well-known appetite-reducing hormone leptin decreases. This signal is transmitted to the hypothalamus, and consequently, appetite increases [36]. However, inflammatory cytokines, such as TNF-α and IL-1, act on this unique regulatory loop and disrupt the body’s response to starvation [37]. Thus, the reduced food intake in our SLC-transplanted rats, observed despite reduced blood leptin levels, suggests that this dysfunction is possibly because of TNF-α and IL-1. Although further investigation into the humoral factors that affect appetite is required, on the basis of our results, we deduce that inflammatory and catabolic factors, such as PTHrP, TNF-α, IL-1β, and FGF21, may have been involved in the development of the severe anorexia observed in the rats in our cancer cachexia model.

In our study, LIF was detected in culture supernatants of SLC cells and tumor extracts of SLC-transplanted rats, and in the rats’ sera at low levels. LIF is a member of the IL-6 superfamily and is characterized by the use of the common receptor chain glycoprotein 130 and activates the Janus kinase–signal transducer and activator of transcription signaling pathway [38]. A recent study showed that mice transplanted with colon 26 cells exhibited LIF-dependent muscle atrophy, despite having a substantially low blood concentration of LIF (approximately 10 pg/mL) [39]. This finding supports that such low levels of LIF were sufficient to induce severe muscle degradation in our cancer cachexia model. 

PTHrP also plays a crucial role in muscle atrophy. Kir et al. demonstrated that PTHrP induces energy expenditure in adipose tissues and atrophy in the skeletal muscle. Neutralization of PTHrP in Lewis lung cancer tumor-bearing mice inhibits the loss of both adipose tissue and skeletal muscle [26]. Moreover, blood PTHrP levels were elevated in a group of patients with advanced cancers with reduced lean body mass and increased energy expenditure [40,41]. Therefore, we speculate that the high PTHrP blood levels in our study greatly contributed to the pathological conditions observed in the SLC tumor-bearing rats. 

Inflammatory cytokines, which we detected in the cytokine array, including IL-6 and TNF-α, are associated with muscle atrophy through the induction of E3 ubiquitin ligase genes and autophagy genes [2,3]. Regarding muscle synthesis, the low IGF-1 blood levels in the rats in our model concurred with the observations reported in other cancer cachexia models [42]. IGF-1 protein activates the phosphoinositide 3-kinase/protein kinase B signal and stimulates protein synthesis through the mammalian target of rapamycin and S6 kinase signaling pathways in skeletal muscle [43]. Furthermore, the IGF-1 protein promotes the phosphorylation of protein kinase B and suppresses the gene expression levels of muscle-specific E3 ligases through downstream phosphorylation of forkhead box class O protein [44]. Thus, we can conclude that the muscle atrophy observed in the rats in our cancer cachexia model was caused by both accelerated muscle degradation and decreased muscle synthesis. 

A notable factor identified by the cytokine array using the blood of the SLC-transplanted rats was NGAL. This is because NGAL was produced in the culture supernatants of the SLC cells at steady state and was also detected at considerable levels in the sera of the SLC-transplanted rats. However, NGAL was barely detectable in the sera of the control rats. Moreover, NGAL was initially identified as a potent bacteriostatic agent, as it blocks ferrous ions to defend against Gram-negative bacteria. NGAL also functions as a stress protein that induces innate immune responses, cell differentiation, tumorigenesis, and cell survival [45]. A study has suggested that the levels of NGAL in the blood are correlated with obesity, insulin resistance, hyperglycemia, coronary heart disease, acute kidney injury, and fatty liver [45]. While NGAL is overexpressed in several cancers, including lung, ovarian, colorectal, and pancreatic cancers, and it is thought to promote cancer cell survival, growth, metastasis, and tumorigenesis, few reports directly correlate NGAL with cancer cachexia [46]. However, most of the pathological characteristics of cancer cachexia we observed in this study are in accordance with the observations in several studies of NGAL. During muscle atrophy, NGAL releases high-mobility group box 1 (a damage-associated molecular pattern) from several cells, which in turn, activates the nuclear factor kappa B pathway and the inflammatory response of the NLRP3 inflammasome. This is essential for the maturation of IL-1β and IL-18 [47]. Furthermore, high-mobility group box 1 contributes to the metabolic reprogramming of the skeletal muscle through the receptor for advanced glycation end products, inducing muscle degradation because of autophagy and releasing free amino acids in the plasma [48]. During anorexia, NGAL crosses the blood–brain barrier and binds directly to melanocortin 4 receptor in hypothalamic neurons. This activates a melanocortin 4 receptor-dependent anorectic pathway that suppresses appetite [49]. Moreover, NGAL acts on erythroid progenitor cells and negatively regulates erythrocyte differentiation and production during hematopoiesis [50]. Although much is still unknown about the direct effects of NGAL and the underlying mechanism in cancer cachexia, it is possible that NGAL was strongly involved in the pathogenesis of our SLC-transplanted cancer cachexia model. Additionally, we observed an increase in the blood levels of PAI-1 in the SLC-transplanted rats; this increase was approximately 5-fold higher than that in the control rats. It has been reported that steroid-induced muscle wasting is attenuated in PAI-1-deficient mice [51]. Moreover, gene expression of PAI-1 in muscle tissues increases significantly in patients with neurogenic muscle atrophy [52]. However, further study is necessary to determine the extent of the involvement of these factors in the pathogenesis of cancer cachexia in our model.

## 5. Conclusions

We developed a novel cancer cachexia rat model by transplanting SLC cells derived from a rat lung adenocarcinoma cell line. The cancer cachexia in this model was similar to the latest consensus definition and its pathology closely resembled the findings in humans. Thus, our novel SLC-transplanted cancer cachexia rat model is a valuable tool which can be used to conduct further research regarding the etiology of cancer cachexia and to develop treatments and supportive care for affected patients.

## Figures and Tables

**Figure 1 biomedicines-11-02824-f001:**
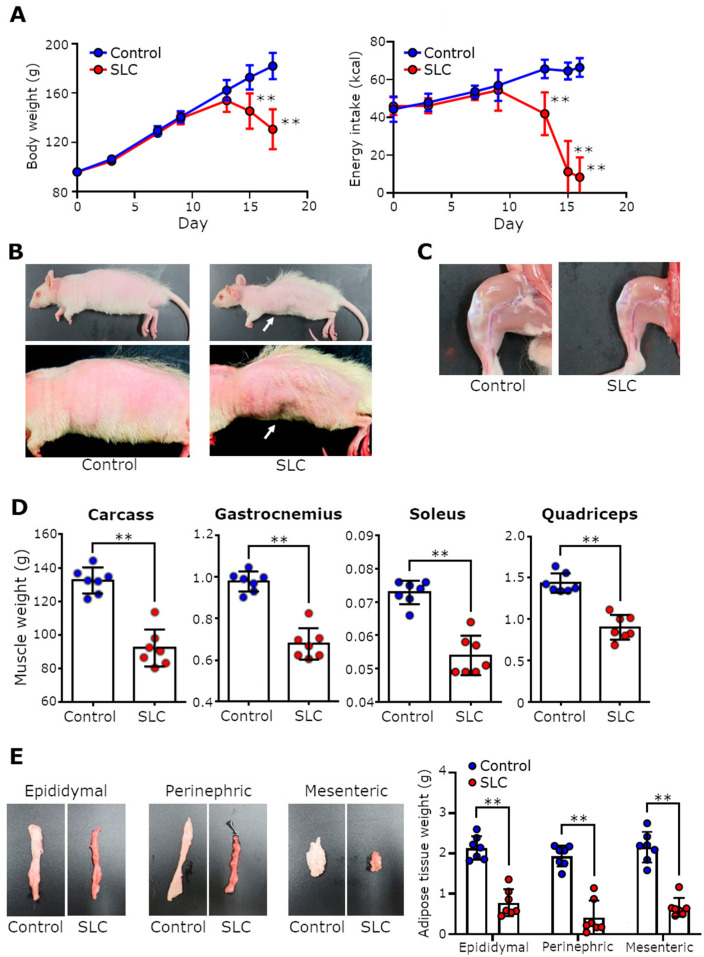
Effects of SLC cell transplantation on the body weight, food intake, muscles, and adipose tissues of rats. (**A**) The body weight and energy intake of control and SLC-transplanted rats over time (*n* = 7/group). Body weight excludes tumor weight. Results are represented as mean ± SD. (**B**,**C**) Representative images of the physical appearance of control and SLC-transplanted rats and their leg muscles, respectively, on day 17 following transplantation. White arrows indicate conspicuous ribs. (**D**) Weight of the carcass, and the gastrocnemius, soleus, and quadriceps femoris muscles of control and SLC-transplanted rats on the day of necropsy. Carcass weight was determined as the weight of the rat body with all organs removed (eviscerated) and with the head retained. (**E**) Representative images of the appearance and mass of the epididymal, perinephric, and mesenteric adipose tissues on day 17 in both control and SLC-transplanted rats. Each point represents the evaluated parameter for one rat; error bars indicate the SD. The statistical significance of each item was determined using Student’s *t*-test. ** *p* < 0.01.; SLC, rat lung adenocarcinoma cell line.

**Figure 2 biomedicines-11-02824-f002:**
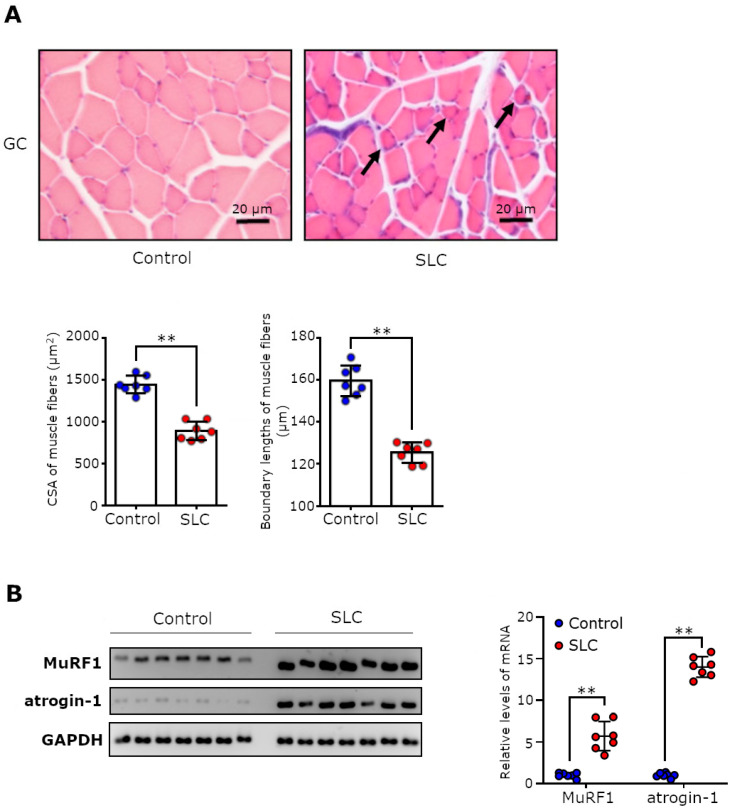
Muscle atrophy and decreased muscle function in SLC-transplanted rats. (**A**) Representative macroscopic photographs of the H&E-stained gastrocnemius muscle, and the CSAs and boundary lengths of the gastrocnemius muscle fibers on day 17 following SLC cell transplantation. Average values of the CSAs and the boundary lengths (five fields of view per sample) were calculated. Black arrows indicate representative muscle fibers with severe atrophy. The scale bar in the photomicrographs indicates 20 µm. (**B**) Optical densities of the MuRF1, atrogin-1, and GAPDH mRNA levels in the rat gastrocnemius muscles and the mRNA levels of MuRF1 and atrogin-1 normalized to that of GAPDH. Each point represents the expression levels in one rat, and error bars indicate the SD. Statistical significance of each parameter was determined using Student’s *t*-test, and the gene expression levels of MuRF1 and atrogin-1 were determined using Welch’s *t*-test. ** *p* < 0.01.; CSA, cross-sectional area; SLC, rat lung adenocarcinoma cell line; GC, gastrocnemius; MuRF1, muscle-specific ring finger protein 1; GAPDH, glyceraldehyde-3-phosphate dehydrogenase.

**Figure 3 biomedicines-11-02824-f003:**
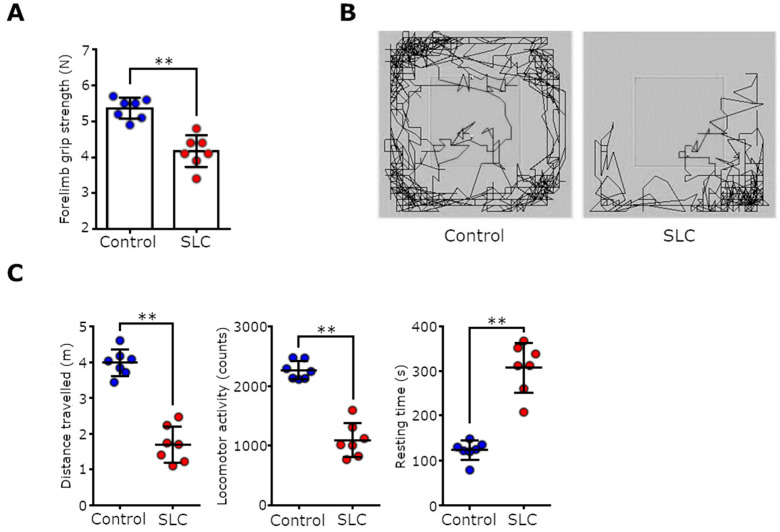
Muscle function and locomotor activities of the SLC-transplanted rats. (**A**) Forelimb grip strengths of the control and SLC-transplanted rats on day 17 following transplantation. (**B**) Typical examples of the trajectories of a control rat and an SLC-transplanted rat in an open field during the first 10 min on day 16 following transplantation. (**C**) Distance, locomotor activity, and resting time calculated by analyzing the open field test data for the control and SLC-transplanted rats. Each point represents the parameters calculated for a single rat, and error bars indicate the SD. Statistical significance for each item was determined using Student’s *t*-test. ** *p* < 0.01.; SLC, rat lung adenocarcinoma cell line.

**Figure 4 biomedicines-11-02824-f004:**
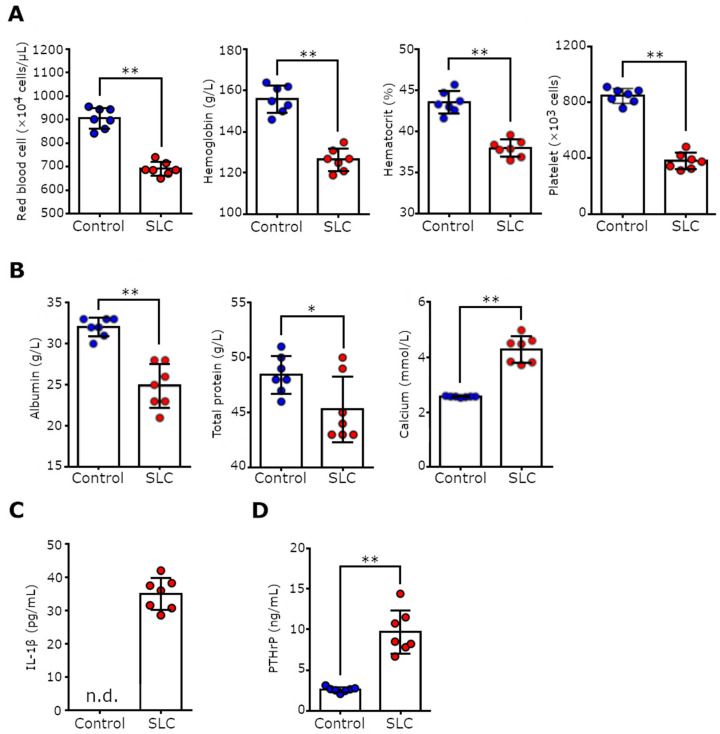
The serum chemistry and hematological profile of the rat model of cancer cachexia. (**A**) The hematological assessment of blood samples from the control and SLC-transplanted rats on day 17 following transplantation. (**B**) Serum chemistry of control and SLC-transplanted rats on day 17 after transplantation. Serum levels of (**C**) IL-1β and (**D**) PTHrP in control and SLC-transplanted rats. Each point represents the evaluated parameter for a single rat, and error bars indicate the SD. Statistical significance of each parameter was determined using Student’s *t*-test, and the blood calcium concentration was determined using Welch’s *t*-test. * *p* < 0.05, ** *p* < 0.01.; n.d., no detection; SLC, rat lung adenocarcinoma cell line; IL-1β, interleukin-1 beta; PTHrP: parathyroid hormone-related peptide.

**Figure 5 biomedicines-11-02824-f005:**
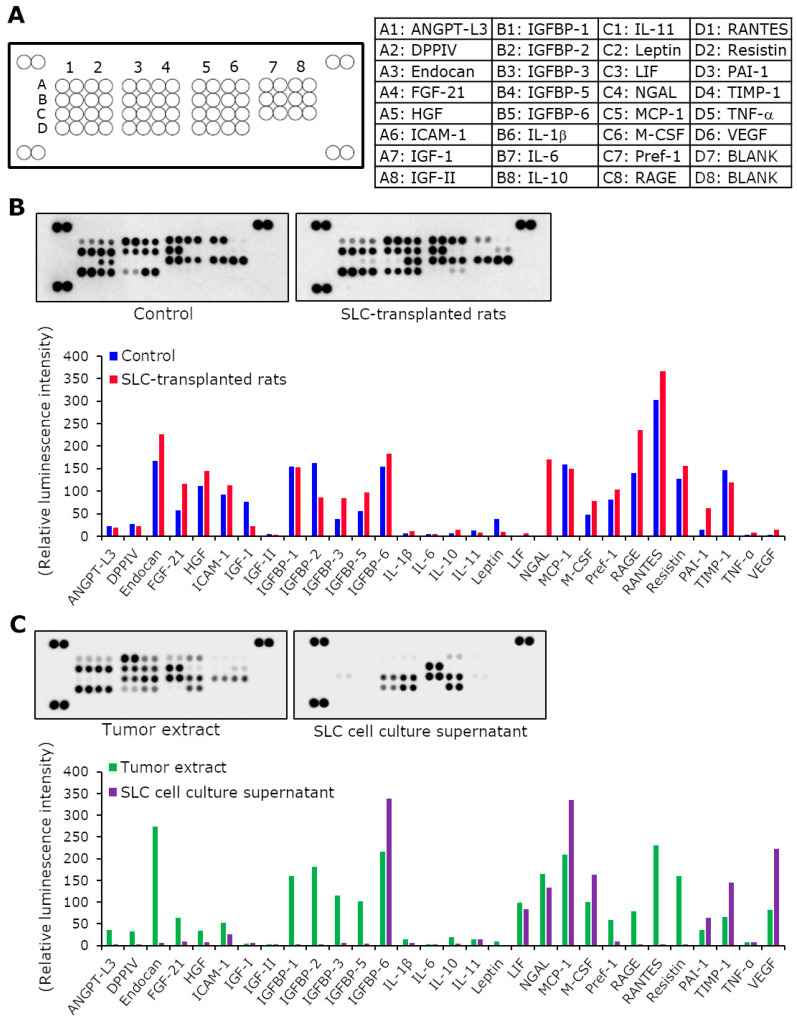
Cytokine production levels in the rat serum, tumor extracts, and cell culture supernatants. (**A**) Cytokine array membrane. Each spot corresponds to the position of a specific cytokine on the membrane. Each cytokine is arranged in duplicate, and the four corners of the membrane represent the control spots. (**B**) Optical densities of the cytokines on the membrane and the graphical representation of the relative luminescence intensity of each cytokine in the pooled sera of control and SLC-transplanted rats. (**C**) Optical densities of cytokines on the membrane and the graphical representation of the relative luminescence intensity of each cytokine in pooled tumor extracts derived from SLC-transplanted rats and in SLC cell culture supernatants. SLC, rat lung adenocarcinoma cell line; ANGPT, angiopoietin-like protein; DPPIV, dipeptidyl peptidase-4; FGF, fibroblast growth factor, HGF, hepatocyte growth factor; IGF, insulin-like growth factor; IGFBP, IGF-binding protein; IL, interleukin; LIF, leukemia inhibitory factor; NGAL, neutrophil gelatinase-associated lipocalin; MCP, monocyte chemoattractant protein; M-CSF, macrophage colony-stimulating factor; M-CSF, macrophage colony-stimulating factor; RAGE, receptor for advanced glycation end products; RANTES, regulated on activation, normal T cell expressed and secreted; PAI, plasminogen activator inhibitor; TIMP, tissue inhibitors of metalloproteinases; TNF, tumor necrosis factor; VEGF, vascular endothelial growth factor.

**Figure 6 biomedicines-11-02824-f006:**
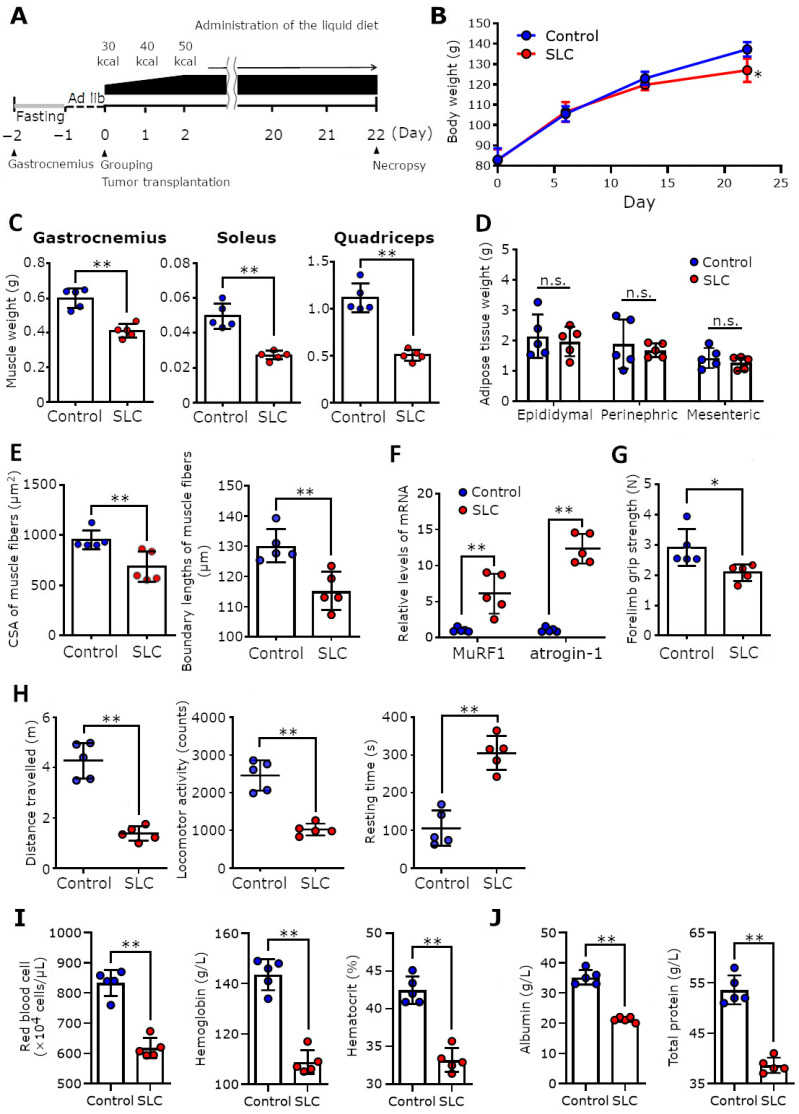
Effects of SLC transplantation on rats subjected to forced tube feeding. (**A**) Schematic representation of the experimental design for SLC transplantation in the force-fed rats. On the day after gastrostomy, the rats had free access to the liquid diet, and on the following day, they were grouped, transplanted with the SLC cells, and fed an isocaloric liquid diet, thereafter. (**B**) These graphs show the body weight of each group over time (*n* = 5/group) (tumor weight was excluded from the body weight). (**C**,**D**) Weight of skeletal muscles and adipose tissues in the control and SLC-transplanted rats measured on the day of necropsy. (**E**) CSAs and boundary lengths of the gastrocnemius muscle fibers in each rat group. Average values of the CSAs and the boundary lengths (five fields of view per sample) were calculated. (**F**) mRNA levels of MuRF1 and atrogin-1 (normalized with that of glyceraldehyde-3-phosphate dehydrogenase in the gastrocnemius muscle in each rat group. (**G**) Forelimb grip strength in each group on day 22. (**H**) Locomotor parameters calculated by analyzing the movement of rats in each group in an open field on day 22. (**I**,**J**) Hematological assessment and serum chemistry of control and SLC-transplanted rats on the day of necropsy. Each point represents a parameter calculated for one rat, and error bars indicate the SD. The statistical significance of each item was determined using Student’s *t*-test, and the mRNA levels of MuRF1 and atrogin-1 were analyzed using Welch’s *t*-test. * *p* < 0.05, ** *p* < 0.01; CSA, cross-sectional area; n.s., not significant; SLC, rat lung adenocarcinoma cell line; MuRF1, muscle-specific ring finger protein 1.

## Data Availability

The data presented in this study are available on request from the corresponding author.

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
