# Peer review of "Development and Characterization of a Cancer Cachexia Rat Model Transplanted with Cells of the Rat Lung Adenocarcinoma Cell Line Sato Lung Cancer (SLC)"

_biomedicines, 2023, doi:10.3390/biomedicines11102824_

Round 1

Reviewer 1 Report

Manuscript has a good title.

English language has good quality. figures have acceptable quality.

There are some modifications that are essential to be exerted in the manuscript.

1. About page 2, line 81, section "2.1. Animals"

+ Please mention the source of permission and also permission ID of your manuscript.

+ Please mention or cite the protocole by which you have performed animal housing

2. About page 2, line 91, section "2.2. Cell culture and tumor inoculation"

+ Please mention or cite the protocole by which you performed cell culture

3. About page 3, line 102, section "2.3. Grip strength test"

You performed this section based on which scientific protocole?

4. Line 122 in page 3

Why you obtain sample from venae cava of rats?

5. Line 152 in page 4

Why you extracted the RNA from the rats skeletal muscle tissues?

6. In line 195-197 in page 4-5

You have mentioned that after 10 days the body weight of rats with SLC was lower than control but in figure 1A, the statistically significant difference between body weight of rats with SLC and control is in day 15 to 20.

Please reform this sentence.

7. Line 197-198 in page 5

Do you think the loose of body weight in SLC rats is because of SLC condition or deduction in food intake? Please explain.

8. About figure 1B

Please show differences between details of this figure by some arrows

9. About figure 2A

+ Please make the scales of this figure bigger so it can be easily visible

+ Please determine some important details of figure by arrows

10. In section "5. Conclusions" in page 15

The authors should conclude from their

findings in this section and present a brief, effective and scientific conclusion based on all of their previous findings. Please mention this conclusion in this section.

11. Please check and adjust the "Reference list" based on the regulations of reference list of journal. (Titles, doi, the name of journal and ... )

Reviewer 2 Report

Cancer cachexia is known to be a complex malnutrition syndrome causing progressive dysfunction. This syndrome is accompanied by losses of protein and energy caused by a decrease in nutrient intake and the development of metabolic disorders. 80% of patients with advanced cancer develop cancer cachexia; however, effective targeted treatments remain to be developed. The study authors developed a new rat model that mimics the human pathology of cancer cachexia to elucidate the mechanism underlying the onset and progression of this syndrome using a rat lung adenocarcinoma cell line. The authors observed that rats transplanted with SLC exhibited severe anorexia, weight loss, muscle atrophy, and weakness. In addition, they had obvious signs of cachexia, such as anemia, inflammation and low serum albumin levels. The rats also experienced weight and muscle loss despite adequate tube feeding.

Overall, I liked the article, it was written competently and logically, and there were no comments on the research design.

I have a question for the authors: how correct is the proposed model if it excludes one of the important factors of cachexia - decreased appetite, since the rats were fed through a tube?

Round 2

Reviewer 1 Report

No more comment.